# Ecological and Human Health Risk Assessment of Heavy Metal Pollution in the Soil of the Ger District in Ulaanbaatar, Mongolia

**DOI:** 10.3390/ijerph17134668

**Published:** 2020-06-29

**Authors:** Enkhchimeg Battsengel, Takehiko Murayama, Keisuke Fukushi, Shigeo Nishikizawa, Sonomdagva Chonokhuu, Altansukh Ochir, Solongo Tsetsgee, Davaadorj Davaasuren

**Affiliations:** 1Department of Transdisciplinary Science and Engineering, School of Environment and Society, Tokyo Institute of Technology, Yokohama, Kanagawa 226-8502, Japan; murayama.t.ac@m.titech.ac.jp (T.M.); nishikizawa.s.ab@m.titech.ac.jp (S.N.); 2Department of Environment and Forest Engineering, School of Engineering and Applied Sciences, National University of Mongolia, Ulaanbaatar 210646, Mongolia; ch_sonomdagva@num.edu.mn (S.C.); altansukh@seas.num.edu.mn (A.O.); 3Division of Global Environmental Science and Engineering, Graduate School of Natural Science and Technology, Kanazawa University, Kanazawa, Ishikawa 920-1192, Japan; fukushi@staff.kanazawa-u.ac.jp (K.F.); osko_8888@yahoo.com (S.T.); 4Laboratory of Environmental Engineering, National University of Mongolia, Ulaanbaatar 210646, Mongolia; 5Department of Geography, School of Art and Sciences, National University of Mongolia, Ulaanbaatar 210646, Mongolia; davaadorjd@gmail.com

**Keywords:** health risk assessment, ecological risk assessment, ger district, heavy metals, soil pollution

## Abstract

The aim of the present study was to evaluate human health and potential ecological risk assessment in the ger district of Ulaanbaatar city, Mongolia. To perform these risk assessments, soil samples were collected based on reference studies that investigated heavy element distribution in soil samples near the ger area in Ulaanbaatar city. In total, 42 soil samples were collected and 26 heavy metals were identified by inductively coupled plasma optical emission spectrometry (ICP-OES) and inductively coupled plasma mass spectrometry (ICP-MS) methods. The measurement results were compared with the reference data in order to validate the soil contamination level. Although there was a large difference between the measurement results of the present and reference data, the general tendency was similar. Soil contamination was assessed by pollution indexes such as geoaccumulation index and enrichment factor. Mo and As were the most enriched elements compared with the other elements. The carcinogenic and noncarcinogenic risks to children exceeded the permissible limits, and for adults, only 12 out of 42 sampling points exceeded the permissible limit of noncarcinogenic effects. According to the results of the ecological risk assessment, Zn and Pb showed from moderate to considerable contamination indexes and high toxicity values for ecological risk of a single element. The Cr and As ranged as very high ecological risk than that of the other measured heavy metals.

## 1. Introduction

Coal is the primary source of energy in Ulaanbaatar, Mongolia, and accounts for approximately 90% of the energy sector in Mongolia [1]. Ulaanbaatar has a population of 1.5 million and more than half of the households (193,529) were located in a ger district in 2019 [2]. As mentioned by Davaabal. B et al. [3], a household of ger residents uses approximately 1.1–1.3 million tons of raw coal a year, which is imported from the coal mines of Baganuur and Nalaikh, because raw coal is a source of energy. As a result, around 198,000–230,000 tons of coal ash are generated and require disposal per year. Unfortunately, due to the lack of management for coal ash disposal in the ger area, residents are disposing of coal ash in illegal and unregulated points such as public waste points and ravines near the ger area. As a result, coal ash is becoming one of the main sources of soil pollution in the ger area of Ulaanbaatar city [4,5,6,7]. The coal ash contains a high concentration of heavy metals compared with other geological materials as the concentration of heavy metals is enriched four to ten times after combustion and is harmful to human health [8,9]. These heavy metals are easily transported by air and humans can easily be affected through ingestion, inhalation, and dermal contact. Moreover, these heavy metals easily contaminate the environment such as in soil and water [10,11,12]. 

Therefore, it is important to assess the risk caused by heavy metals to protect human health and the environment. As defined by the United States Environmental Protection Agency (USEPA) [13], the risk level can be defined by three parameters: first, the amount of chemical substances such as heavy metals in the medium; second, the frequency of humans and receptor exposure; and third, toxicity quality of chemicals. Risk assessments are generally conducted in areas where chemical contamination can results in health risks to human and ecological receptors. Therefore, it is necessary to conduct human health and potential ecological risk in the ger area under the aforementioned circumstances. It is necessary to investigate the concentration of heavy metals and contamination degree of soil in order to perform human health and ecological risk assessments.

The soil contamination, caused by heavy metals, of Ulaanbaatar city and its distribution has been investigated for the last decade [3,4,5,6,7,14,15]. For instance, Batjargal. Ts et al. [5] investigated the heavy metal pollution of 22 samples in the soil of Ulaanbaatar and determined the mean concentrations of As (arsenic), Cd (cadmium), Cr (chromium), Cu (copper), Ni (nickel), Pb (lead), and Zn (zinc). The study showed that the concentration of the heavy metals did not exceed the Mongolian National Standard (MNS 5850:2008), except for element As [16]. The arsenic was only an element that exceeded the MNS standard. It was already stated in Davaabal. B et al. [3] that the concentration of As in the coal ash of the Baganuur and Naliakh mines measured over 20 times higher than the MNS standard. The collected surface soil samples from 285 locations of Ulaanbaatar city. The concentrations of the Pb, Cr, Ni, Zn Cd, and Co were measured by atomic absorption spectrometry (AAS). The author highlighted that the concentration of Cr, Pb, and Zn exceeded MNS standards at some locations such as waste points in the ger area, some auto markets, and leather factory. Surface soil samples from 136 locations around Ulaanbaatar city were collected by the Urban Environmental Agency in Mongolia [15] and 27 heavy metals were identified by the x-ray fluorescence method. The measurement results revealed that the concentration of As was higher at waste points of the ger area, automakers, and glass factory in the Nalaikh district. The concentration of As, Cr, Pb, Zn, and Ni were identified by the AAS technique in 27 surface soil samples [7]. The average concentration of 27 samples for As measured was 4.6 times higher than the MNS standard at Ulaan-chuluut. The concentration of Pb was higher than the MNS standard in the sample collected from Dari-ekh and Ulaan-chuluut. Moreover, human health risk assessment was performed by the USEPA method in [14] by using previously published data mentioned by Sonomdagva. Ch et al. [7]. According to the health risk assessment, the risks caused by ingestion, dermal contact, and inhalation pathways were different for children and adult. The non-carcinogenic risk was higher for children when compared with adults. There was no carcinogenic risk from As, Cr, and Ni. Although there was no risk from As, Cr, and Ni mentioned by Sonomdagva.Ch et al. [14], it can be observed that the soil is contaminated by arsenic (As), based on the findings of all the aforementioned studies [3,14,15]. 

A limited number of studies have been performed for human health risk assessment for the public who live in the ger area in Ulaanbaatar, while there has been no ecological risk assessment in this area. Therefore, to fill this gap, the main purpose of the present study was to perform human health and ecological risk assessment in detail. To achieve this purpose, first, literature materials on heavy metal concentration were collected. Using the collected data, it was decided to retake soil samples from highly-contaminated points based on the literature data. Second, the concentration of heavy metals in the retaken soil samples was identified and compared with the literature data. Third, the identified concentration in the present study was further used to assess both human health and ecological risks.

## 2. Methodology

### 2.1. Study Area

Mongolia is located in East Asia and bordered by China and Russia, as illustrated in Figure 1. Ulaanbaatar is the capital city of Mongolia. The study area placed in the Ulaanbaatar and geographic coordinates of the research area were from 47°54′28.04″ N to 47°55′45.6″ N latitude and from 106°34′06.27″ E to 107°00′07.4″ E longitude. 

### 2.2. Selection of Sampling Locations and Sampling

Although a number of studies have been performed to identify the concentration of heavy metals in soil samples as mentioned in the introduction, three main studies were selected in the present study as most of soil samples in these three studies [6,14,15] were collected from the ger area of Ulaanbaatar city and showed a high concentration of heavy metals. In these studies, there were 119 soil samples in total, as illustrated in Figure 2. 

Based on these data, it was decided to recollect soil samples from 42 points that showed a high concentration and exceeded the Mongolian National Standard, especially for As. The soil samples were recollected at a depth of 10–30 cm using a plastic spatula and stored in self-locking polyethylene bags; the samples were collected between May and June 2019. The locations of the soil samples were recorded using a handheld Garmin Global Positioning System. The soil samples were dried at room temperature (20 °C) for three days at the National University of Mongolia. After the soil samples were dried, they were transported to Japan. The soil samples were stored for a week in a desiccator with airtight plastic bags and used for chemical analysis in the Kanazawa University laboratory.

### 2.3. Laboratory Experiment at Kanazawa University 

The soil samples were dried in the National University of Mongolia laboratory after the removal of stone and fragments. The dried samples were sieved through a 0.25-mm sieve and crushed until powder. The reagent used the analytical method and was divided into five consecutive steps. In the first step, solid samples of 0.05 g each was digested completely in 3 mL 60% HNO_3_ with 3 mL 48% hydrofluoric acid (HF) and heated at 120 °C for 48 h. In the second step, 3 mL of 30 % hydrochloric acid (HCl) was added and heated at 120 °C for 24 h until dry. In the third step, 10 mL of 0.6% HNO_3_ was extracted and mixed for 24 h in the mix-rotor [17]. The obtained extraction solutions were filtered to the I-boy through a 0.20-µm cellulose membrane filter in the fourth step. In the final step, the sample was diluted 50 times by using a 0.6 % HNO_3_ solution for the ICP-MS and ICP-OES measurements. The procedure matrixes are summarized in Table 1. BCR (Community Bureau of Literature) was a modified extraction procedure for the analysis of heavy metals in soil [18]. 

The experiments were performed using inductively coupled plasma optical emission spectrometry (ICP-OES, Varian 710-ES) and inductively coupled plasma mass spectrometry (ICP-MS, X-Series) at the Department of Global Environmental Science and Engineering at the Graduate School of Natural Science and Technology of Kanazawa University, Japan. The calibration curve was prepared with known concentrations of each element based on the multi- and single-standard solution. Low-level heavy metal (20 elements) concentrations for Cr, Co, Cu, As, Se, Cd, Pb, Mo, Zn, Al, V, Kr, Rb, Sr, Ag, Cs, Ba, Bi, Th, and U were analyzed using high sensitivity equipment of ICP-MS. Common metal concentrations (six elements) for Ca, Fe, K, Mg, Na, and Mn were analyzed using high ICP-OES. The detection limit depends on a heavy metal concentration around 50–100 ng/kg for low-level elements. Moreover, it should be noted that these elements have been listed on the Priority Pollutant list by the USEPA due to their potential toxic characteristics. They have also attracted increasing attention worldwide [19,20]. In addition, background and MNS were compared in the concentration of heavy metals in the soil for all of the samples.

### 2.4. Pollution Indices of Soil Contamination

To assess soil heavy metal contamination, pollution indices such as enrichment factor (EF) and geo-accumulation index (I_geo_) were considered. The EF was used to assess the degree of soil contamination while the I_geo_ was used to assess the potential anthropogenic impact and the background level of natural fluctuations [21,22]. The nine out of 26 elements were calculated for pollution indexes due to the reference data available. The EF and I_geo_ are estimated as Equation (1) and Equation (2), respectively.
(1)$$\mathrm{EF}=\frac{{\left(\frac{M}{\mathrm{Fe}}\right)}_{\mathrm{sample}}}{{\left(\frac{M}{\mathrm{Fe}}\right)}_{\mathrm{background}}}$$
where ${\left(\frac{M}{Fe}\right)}_{sample}$ denotes the concentration ratio between the heavy metal (M) and Fe in the sample and ${\left(\frac{M}{Fe}\right)}_{background}$ denotes the background concentration ratio between the heavy metal (M) and Fe in the sample;
(2)$$\mathrm{Igeo}={\mathrm{Log}}_{2}\left(\frac{C_{n}}{1.5\cdot B_{n}}\right)$$
where C_n_ is the measured concentration of element *n* = 6; B_n_ denotes the value of the background concentration; these values were selected from [23]. Cr = 45, Co = 18, Cu = 25, As = 12, Cd = 1.0, Pb=20, Mo = 1.9, Mn = 710, and Zn = 60.

The constant 1.5 was considered due to the possible changes in background data due to lithological variations. EF and I_geo_ were categorized into six and seven classes, respectively, as listed in Table 2.

### 2.5. Potential Human Health Risk Assessment

Potential health risks of the residents living in the ger area of Ulaanbaatar city was performed by the USEPA method [24,25]. The residents living in the ger area of Ulaanbaatar city can be exposed to the contaminated soil through ingestion, and dermal and inhalation pathways. The exposure pathways for every heavy metal in the soil can be expressed by the average daily intake (ADI) according to Equations (3)–(5).
(3)$$ADI_{Ingestion}=\frac{C\;\times \;IngR\;\times \;EF\;\times \;ED\;\times \;CF}{BW\;\times \;AT}$$
(4)$$ADI_{dermal\;contact}=\frac{C\;\times \;SA\;\times \;FE\;\times \;ABS\;\times \;EF\;\times \;ED\;\times \;CF}{BW\;\times \;AT}$$
(5)$$ADI_{inhalation}=\frac{C\;\times \;InhR\;\times \;EF\;\times \;ED}{PEF\;\times \;BW\;\times \;AT}$$

Noncarcinogenic risk or hazard index (HI) is estimated by the ADI and reference dose (RfD) [25] while carcinogenic risk or cancer risk (CR) is estimated by the ADI and slope factor (SF) [25]. Eight out of 26 elements were used to assess for noncarcinogenics, while only four elements were used to assess for carcinogenics because SF and RD were available. The RfD and SF values are listed in Table 3. 

Used equations for noncarcinogenic and carcinogenic risks are shown in Equations (6) and (7) and Equations (8), respectively. The non-carcinogenic risk for a single heavy metal was determined as the hazard quotient (HQ). To find out the total cancer risk, according to Equation (8), the estimated risk caused by every heavy metal for the three pathways were added. In the present study, an permissible limit for the HI was set as 1, while the permissible limit for CR was set as 10^−4^ [28].
(6)$$\mathrm{HQ}=\frac{\mathrm{ADI}}{\mathrm{RfD}}$$
(7)$$\mathrm{HI}=\sum \left({\mathrm{HQ}}_{\mathrm{Ingestion}}+{\mathrm{HQ}}_{\mathrm{Inhalation}}+{\mathrm{HQ}}_{\mathrm{dermal}}\right)=\sum \frac{{\mathrm{ADI}}_{i}}{{\mathrm{RfD}}_{i}}$$
(8)$$\mathrm{TCR}=\sum \left({\mathrm{CR}}_{\mathrm{Ingestion}}+{\mathrm{CR}}_{\mathrm{Inhalation}}+{\mathrm{CR}}_{\mathrm{dermal}}\right)=\sum \mathrm{ADI}\;\times \;\mathrm{SF}\;$$

In other words, if the calculated values exceed the permissible limit values, it is considered as exceeding the permissible limit or it can harmful to human health. The parameters used in the health risk assessment are listed in Table 4. 

### 2.6. Potential Ecological Risk Assessment

The potential ecological risk of heavy metals was developed by Hakason’s model in the present study [34]. According to this model, the contamination index ($CL_{i}$) can be calculated by Equation (9): (9)$$CL_{i}=\frac{C_{i}}{RC_{i}}$$
where
$CL_{i}$Contamination index of heavy metal$C_{i}$Measured concentration of heavy metal in the present study $RC_{i}$Literature concentration of heavy metal in soil sample

The contamination index enabled us to assess the soil contamination and potential ecological risk. This index is estimated by the ratio between the pre-industrial and current measured concentrations of the heavy metal in the soil sample. Six out of 26 elements were used to assess the potential ecological risk to the environment as the toxicity response factor was available for these heavy metal. The concentrations of heavy metals was collected from the literature and are listed in Table 5.

The contamination index was categorized according to the contamination levels as listed in Table 6 [35].

By using the information on the contamination index, the potential ecological risk index of a single element $\left(eRP_{i}\right)$ can be calculated by Equation (10): (10)$$eRP_{i}=TRF_{i}\times CL_{i}$$
where
$eRP_{i}$Ecological risk potential of i-th element in soil sample$TRF_{i}$Toxicity response factor of heavy metal (TRF)

TRF-Pb = 5, Cd = 30, As = 10, Cu = 5, Zn = 1 [36], and Cr = 2 [37]. Finally, comprehensive potential ecological risk (ERP) was estimated by Equation (11):(11)$$PER=\overset{n}{\underset{i=1}{\sum }}eRP_{i}$$

The relation between the potential ecological risk index of a single element $\left(eRP_{i}\right)$, potential ecological risk (ERP), and pollution level are listed in Table 7. 

## 3. Results and Discussion

### 3.1. Measurement Resulst of Heavy Metal Concentrations by ICP-OES and ICP-MS Methods

The measurement results of the heavy metal concentrations in the soil sample by using the ICP-OES and ICP-MS methods are described in this subsection. As mentioned in Section 2.4, the experiment was performed at Kanazawa University. In total, 26 heavy metals in the soil samples were identified by the ICP-OES and ICP-MS methods. The mean concentrations of the elements were found to be in the following sequence: Al > Ba > Sr > Zn > Rb > V > Mo > Pb > Mg > Cr > Cu > Fe > K > As > Ca > Na > Th > Co > Cs > U > Ag > Bi > Mn > Kr > Cd > Se. The measurement results are listed in Table 8 with the Mongolian National Standard (MNS). 

The Zn did not exceed the MNS limits, however, the Zn concentration in sample No 18 (waste point in Chingeltei district) was 384 mg/kg, which exceeds the MNS limits. The concentration of Mo was determined to be higher than the MNS in all of the soil samples sites. In particular, sample No 6 (waste area of the glass factory in Nalaikh district) and 22 (near a ravine in Chingeltei district) were determined to be 219 mg/kg and 334 mg/kg, respectively. The concentration of As was determined to be higher than the MNS limit in all of the soil samples. In particular, sample No 6 was determined to have the highest concentration of As (526 mg/kg), which may be due to the waste area of the glass factory in Nalaikh district. The concentration of other elements was determined to be lower than the MNS. 

### 3.2. Comparison between Present and Literature Data

In the present study, 42 soil samples were recollected based on the literature data, as mentioned in Section 2.2. The collected samples were analyzed by using the ICP-OES and ICP-MS methods and 26 heavy metals were identified in the present study. As the location of the soil samples in the present study and literature were the same, the concentration of both studies was compared to assess the difference of the contamination level. The literature studies identified a limited number of elements, therefore only possible elements were compared with the data of the present study. The detailed explanations for each element are given in this section. It should be highlighted that measurement methods used in the literature were atomic absorption spectrometry (AAS) and x-ray fluorescence (XRF). The measurement results of the present study were compared with previous literature to assess the difference between the present and previous studies, although the compared measurement results were different due to many parameters such as measurement condition and difference in measurement technique. 

Arsenic (As): Measurement result of element arsenic (As) in the present study is illustrated in Figure 3a and the measurement data were compared with previous literature and the MNS values. In Figure 3a, the horizontal and vertical axes represent the concentration measured by the present and literature studies, respectively. The red line represents the Mongolian National Standard. If concentration values exist inside the red lines, it means that the concentration is under the standard value. The measurement results of the present study revealed that all samples exceeded the MNS value for the present and previous literature. In Figure 3a, the dotted black (20 mg/kg) and pink (40 mg/kg) lines represent the deviation from the centerline where two measurement values are equal. The difference between the present and literature data for As was under 40 mg/kg, as illustrated in Figure 3a. The measured concentration of arsenic (As) in sample No 6 was extremely high when compared with the other samples in both the previous literature and present studies. The sample No 6 was taken from a waste area of the old glass factory that is in the ger area of the Nalaikh district, Ulaanbaatar, Mongolia. The glass is made of a molten mixture of heavy metals including As. Therefore, the concentration of As in the sample No 6 might be measured as extremely high (526 mg/kg of As). 

Chromium (Cr): The measurement result of element chromium (Cr) in the present study is illustrated in Figure 3b and the measurement data were compared with the previous literature and MNS values. According to the previous study, 41 out of 42 samples were lower than the MNS standard, except for sample No 13. Although the measurement results are scattered in Figure 3b, the concentration of element Cr in the present study revealed that all measurement data were lower than the MNS standard.

Lead (Pb): The measurement result of element lead (Pb) in the present study is shown in Figure 3c and the measurement data were compared with the previous literature and MNS values. According to the previous study, 41 out of 42 samples were lower than the MNS standard, except for sample No 35. On the other hand, the measurement results of element Pb in the present study revealed that all measurement data were lower than the MNS standard.

Zinc (Zn): The measurement result of element zinc (Zn) in the present study is shown in Figure 3d and the measurement data were compared with the previous literature and MNS values. According to the previous study, 41 out of 42 samples were lower than the MNS standard, except for sample 7. On the other hand, the measurement result of element Zn in the present study revealed that all measurement data were lower than the MNS standard, except for sample No 18. An excess of Zn can affect inhalation, irritating the nose and throat, and also cause wheezing and coughing. Additionally, it can affect the male reproductive system and decrease sperm count.

Copper (Cu): The measurement results of element copper (Cu) in the present study is shown in Figure 3e and the measurement data were compared with the previous literature and the MNS values. According to the previous study, 41 out of 42 samples were lower than the MNS standard, except for sample No 3. On the other hand, the measurement results of element Cu in the present study revealed that all measurement data were lower than the MNS standard.

Cadmium (Cd): Although Cd was identified in the present study, there were only measurement data for samples No. 38, No. 41, and No. 42 in the previous literature. Therefore, the comparison was made at only those points. However, measurement data of the previous and present studies did not exceed the MNS value and there were no significant differences.

### 3.3. Result of Pollution Indices of Soil Contamination

Estimated EF and Igeo using the concentration of heavy metals is shown in Figure 4 and Figure 5, respectively. In Figure 4 and Figure 5, the x-axis represents the number of elements while the y-axis represents the heavy metals identified in the present study.

As shown in Figure 4, there was no enrichment for element Cr, Co, Cu, Zn, Cd, Pb, and Mn. Although EF value for 41 out of 42 samples was estimated to be lower than 2, and the highest EF value was estimated as 8.5 for As at sample No 6. Mo was the most enriched element in a comparison with other elements as illustrated in Figure 4. The highest and lowest EF values for element Mo was estimated as 22.4 at sample No 6 and 0.8 at sample No 31, respectively. Samples No 6 and No 22 were categorized as severe enriched for element Mo. Mo has a potential anthropogenic contamination source and is enriched in coal combustion residues [38].

As shown in Figure 5, the Igeo values were estimated as moderately contaminated for most of sample points for Mo (please see the range of the color bar next to the image). The maximum value of Igeo was estimated as 2 at sample No 22 for Mo. The measurement result revealed that there was no contaminated soil sample with Co and Mn. The highest value of Igeo for As was estimated at sample No 6. The Cr, Pb, Cd, Zn, and Cu elements were assessed to range from uncontaminated to moderately contaminated. 

### 3.4. Result of Health Risk Assessment

#### 3.4.1. Noncarcinogenic Risk Assessment 

The noncarcinogenic hazards and the Hazard Quotient of all elements were not evaluated in the case of some exposure pathways, namely the inhalation and dermal pathways for Co and Mo. This was because the noncarcinogenic RfD for these pathways was unavailable. Table 9 and Table 10 show the quantitative value of HI for each element and pathway for the child and adult.

The children were at risk of noncarcinogenic effects, especially through the dermal pathway, which posed the greatest noncarcinogenic risks, followed by the ingestion pathway. Inhalation posed the lowest risk. In the case of children, the three different exposure pathways resulted in the following sequence for the HI of all the metals studied: Co > Mo > Zn > Cu > Cd > Cr > Pb > As. The dermal pathway yielded HQ and HI values greater than 1. 

In the case of adults, the same sequence was determined. However, 12 out of 42 sample sites were assessed to be higher than 1, while other sample sites were assessed at no risk. In particular, Cr, Pb, and As were assessed as the most for children.

It is acceptable that there is no safe level of lead exposure, particularly for children. Pb can lead to brain swelling, kidney disease, cardiovascular problems, nervous system damage, and even death [25]. Therefore all of the sample sites were assessed for chronic exposure, especially for children in the mentioned elements. Results of the noncarcinogenic risk or hazard index (HI) for children and adults are shown in Figure 6a,b. 

As illustrated in Figure 6a, the HI values for children were estimated as higher than the threshold, as mentioned in Section 2.5. The dotted red line in Figure 6 represents the threshold or maximum value to accept. The HI values for children were estimated to be higher than the threshold in all soil samples, with the extremely high HI value estimated at sample No 6, as shown in Figure 6a. On the other hand, the HI values were estimated to be lower than the threshold for adults, except for samples No 6, No 16, and No 32. At sample No 6, the highest HI values for adults was estimated to be the same as the children. 

#### 3.4.2. Carcinogenic Risk Assessment 

Slope factor (SF) is used to estimate the carcinogenic risk, as mentioned in Section 2.5. The carcinogenic risk was estimated for Cr, Cd, Pb, and As in the present study. The children were at risk of carcinogenic effects, especially through the dermal pathway, which posed the greatest carcinogenic risks, followed by the ingestion pathway with the same noncarcinogenic risk. Exposure to Cr through the dermal pathway was determined to be the highest carcinogenic risk factor in both children and adults. Moreover, exposure to As through dermal contact was estimated to exceed the permissible limit for both adults and children. This means that residents are still affected by exposure to dose by dose of As. Arsenic causes skin cancer via chronic oral ingestion, either manifested as squamous or basal cell carcinomas. Some evidence suggests that oral ingestion of arsenic may also contribute to lung cancer as well as cancers of the bladder, kidney, liver, and colon. Epidemiological studies indicate that there is an increased respiratory cancer risk from occupational exposure to chromium. Cr can cause stomach and intestinal ulcers, anemia, and stomach cancer. Frequent inhalation can cause asthma, wheezing, and lung cancer. 

The carcinogenic test determined that As and Cr could contribute to the estimated carcinogenic risk imposed on the ger district. In the case of both adults and children, all the elements posing carcinogenic risks in the ger district were determined to exceed the permissible limit at all the soil sampling sites. In particular, the dermal pathway was determined to be the exposure pathway most likely to affect human health in the study areas. According to the results of the health risk assessment, both adults and children in the study area are highly susceptible to the exposure of environmental contaminants due to their living conditions and lifestyles. Major sources of chromium released into to the soil in the ger area are from the disposal of commercial products that contain chromium as well as coal ash. Ger residents have a greater chance of exposure because they live near a waste site that contains coal ash. 

The calculated risks for each element are listed in Table 11 and Table 12 for children and adults, respectively. 

The result of carcinogenic risk or cancer risk (CR) is illustrated for both children and adults in Figure 7a,b.

The dotted red line in Figure 7 represents the threshold or maximum value to accept. The cancer risk values for both children and adult was estimated to be higher than the threshold at all soil samples, and an extremely high HI value was estimated at sample No 6 as shown in Figure 7a,b. As and Cr are major contributions to carcinogenic risk as aforementioned. The ger residents are exposed to these elements by touching the soil and digging or playing with the soil. Children may eat and breathe the dust of soil that contains As and Cr while playing. Dust can be brought into the ger from outside. Moreover, drinking water contamination by natural sources of arsenic and chromium are another possibility [39,40].

### 3.5. Result of Ecological Risk Assessment

As described in Section 3.5, the potential ecological risk caused by heavy metals was estimated by Hakason’s model. In this section, the results of each parameter included in the ecological risk assessment will be described.

#### 3.5.1. Results of Contamination Index

The contamination index of heavy metals for each sample was calculated and the results are shown in Figure 8. The estimated minimum, maximum, and mean values of the contamination index are shown for each heavy metal. The highest contamination index was calculated at sample 18 and sample No 6 for Zn and As, respectively. There were three samples where the contamination index for Pb was estimated as very high risk, while the contamination index for Zn were estimated as very high risk at sample No 22, sample No 27, and sample No 35. The degree of contamination level of heavy metals showed the following sequence As > Zn > Pb > Cu > Cr > Cd.

The results of the contamination index for heavy metal are shown in Table 13 and Figure 8. 

#### 3.5.2. Potential Ecological Risk Index of a Single Element $\left({\mathrm{eRP}}_{i}\right)$

Ecological risk is represented to evaluate the adverse ecological effects occurring as a result of exposure to soil contamination stressors. The ecological risk results are shown in Table 14 and Figure 9. 

It was shown that Cd and Cu were estimated to be lower than 40 and the Zn values were between 54 and 383 in the high toxicity values. The highest value of As was obtained at sample No 6. The ecological risk values for Zn showed moderate and considerable values with a higher toxicity coefficient. Cr and As presented very high ecological risk compared to the other measured heavy metals. Figure 10 illustrates the distribution of potential ecological risk (ERP). 

## 4. Conclusions

When coal is burnt in the stove of a ger dwelling, a lot of ash is produced. A small part of coal ash is dispersed in the air (fly ash) and most will stay in the stove [41]. After the burning process is finished, the remaining ash is collected in a bucket, and mostly thrown into the open street on bare soil. Surprisingly, hot ashes can help reduce a slippery surface and provide cover in the open street in the winter period (November to March) [3]. This is the reason why coal ash and its heavy metals fly in the air as well as accumulate in the soil and dissolve in water.

In the present study, potential human health and ecological risk assessments were performed based on the concentration of heavy metals in the soil samples collected from the ger area of Ulaanbaatar city, Mongolia. In total, 28 heavy metals were identified by the ICP-OES and ICP-MS methods. The measurement results were compared with reference data in order to validate the soil contamination level. Although there was a large difference between the measurement results of the present and reference data, the general tendency was similar. For instance, the measurement results of As concentrations for the present and reference studies both exceeded the Mongolian National Standard at all locations where the sample was taken. Except for element As, the concentration of other elements were under the Mongolian National Standard. The concentration of As in sample No 6 could be measured as extremely high because this sample was collected from the waste area of the old glass factory of the Nalaikh district. The glass is made of a molten mixture of heavy metals including As [42]. The measurement results of the present and reference studies were the same at sample No 6. Moreover, the concentration of Mo was determined as higher than MNS for all soil samples in the present study. In particular, samples No 6 and No 22 were determined as 219 mg/kg and 334 mg/kg, respectively. Therefore, it is necessary to reduce the high concentration of Mo and As using some remediation methods. 

Soil pollution was assessed by pollution indices such as enrichment factor (EF) and geo-accumulation index (I_goe_). The As and Mo sourced were contaminated by the anthropogenic effect. Other elements were assessed with no enrichment and no contamination. 

The carcinogenic risk was estimated to exceed the permissible limit for both adults and children at all of the sample sites. The total noncarcinogenic risk estimated exceeded the permissible limit or exceeded 1 for children at all of the sample sites. For adults, only 12 sample sites were estimated to exceed the permissible limit while the other sample sites were estimated to have no health risk. Pb, Cr, and As can pose serious concerns regarding the potential occurrence of health hazards. 

The degree of heavy metal contamination was increased as follows: Cd > Cr > Cu > Pb > As > Zn. The ecological risk values for Zn and Pd showed moderate and considerable values with a higher toxicity coefficient. Cr and As were in the range of very high ecological risk compared to the other measured heavy metals.

This study found that people residing in the ger district of Ulaanbaatar were found to be at the greatest risk of exposure to heavy metals through the dermal pathway due to contaminated media near waste points, ravine, streets, and auto services in the ger district. Additionally, there was one case of ingestion. The domestic cattle of neighbor nomads usually come to seek food from open dump areas, which may be one source of ingestion pathway [43]. If in the milk of these cattle and meat sales to consumers, it would be necessary to control the products from these cattle by the professional inspection agency of the urban city of Ulaanbaatar. 

Therefore, it is necessary to conduct a risk assessment of the drinking water in ger districts because targeted elements of Cr, Mo, Pb, and As can potentially affect the groundwater in these ger districts. The results of our study are expected to assist in future monitoring of pollution caused by heavy metals as well as the development of environmental standards in Ulaanbaatar, Mongolia. These results will also support the implementation of public policies aimed at ensuring the sustainability of development activities in ger districts. On the other hand, the background value of arsenic in Ulaanbaatar is two times higher than that stated in the Mongolian National Standard, which is less than the background value. This should be taken into account.

Monitoring the concentrations of heavy metals in the soil surrounding the ger district is essential for controlling soil pollution and protecting the residents from the risks posed by heavy metal contamination.

## Figures and Tables

**Figure 1 ijerph-17-04668-f001:**
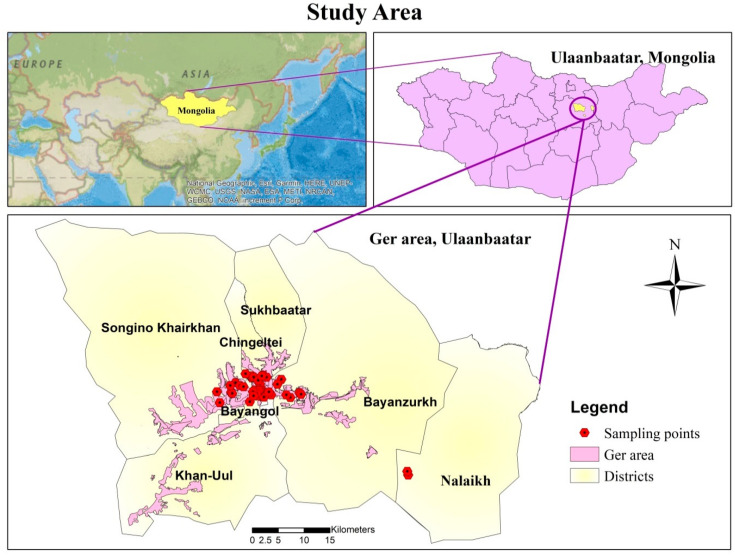
Location of sampling points in the ger district, Ulaanbaatar, Mongolia.

**Figure 2 ijerph-17-04668-f002:**
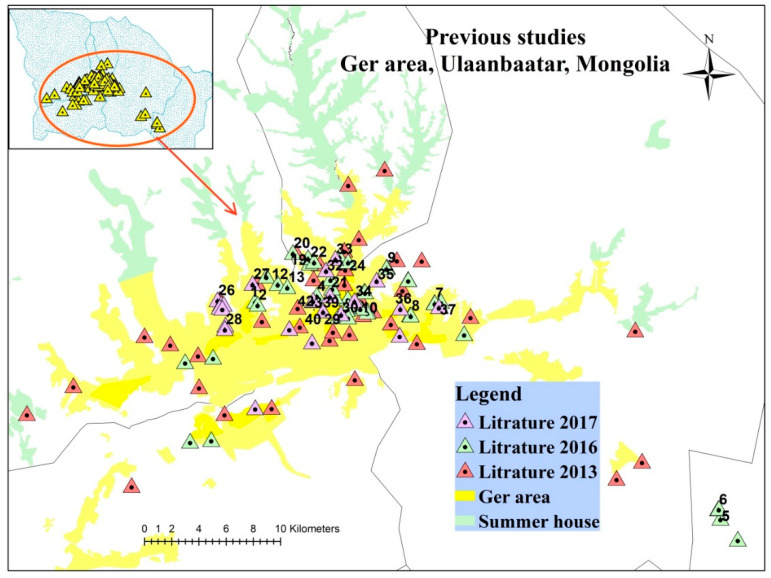
Location of sites in the previous studies and the selected 42 sites.

**Figure 3 ijerph-17-04668-f003:**
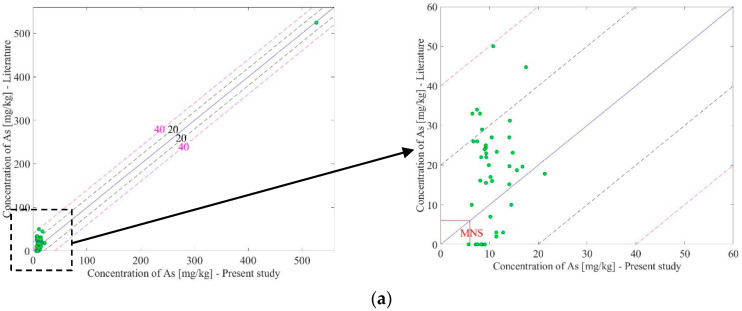
Comparison between the present and literature data on heavy metal concentration. (**a**) Concentration of As; (**b**) Concentration of Cr; (**c**) Concentration of Pb; (**d**) Concentration of Zn; (**e**) Concentration of Cu.

**Figure 4 ijerph-17-04668-f004:**
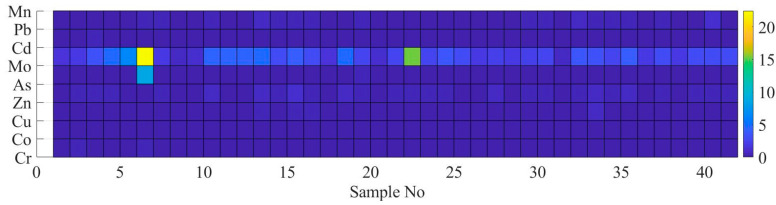
Results of enrichment factor (EF) for heavy metals in each sample.

**Figure 5 ijerph-17-04668-f005:**
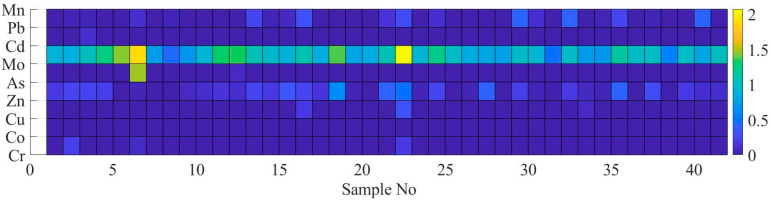
Results of geo-accumulation factor for heavy metals in each samples.

**Figure 6 ijerph-17-04668-f006:**
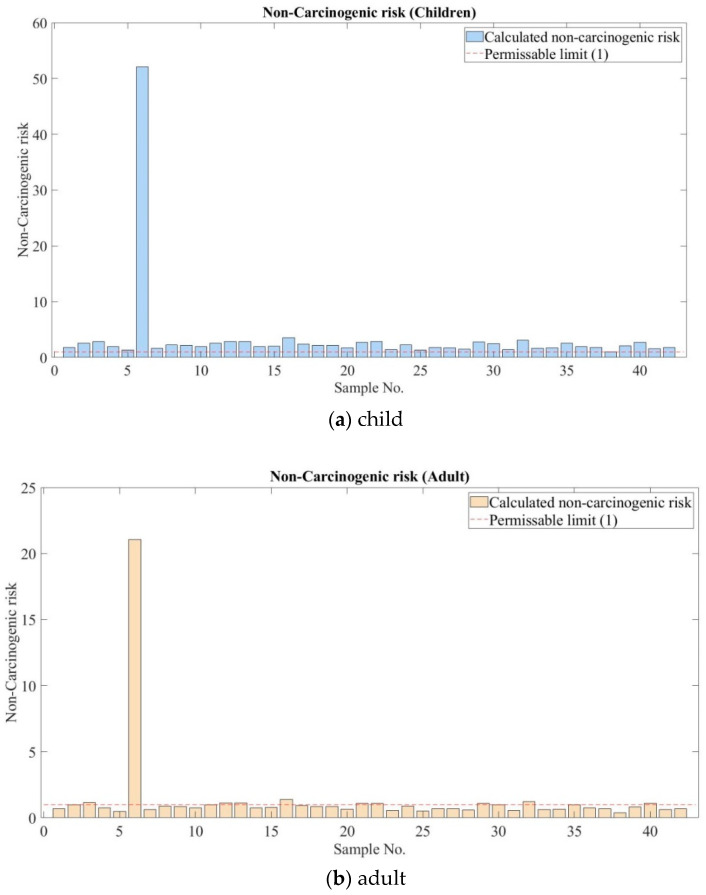
Hazard index (HI) for children and adults living in the ger area.

**Figure 7 ijerph-17-04668-f007:**
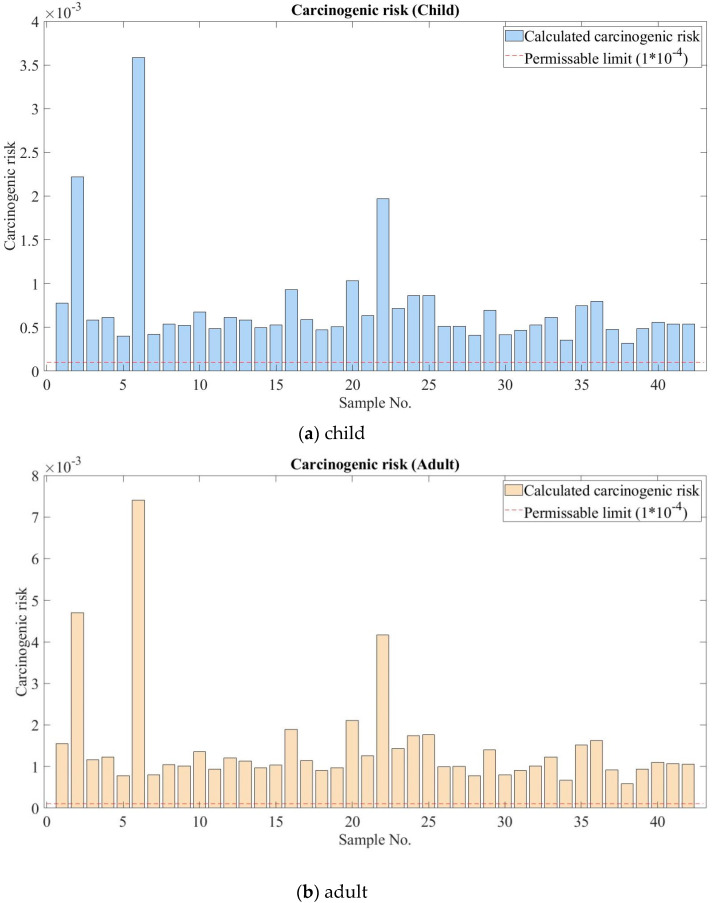
Cancer risk for children and adults living in the ger area.

**Figure 8 ijerph-17-04668-f008:**
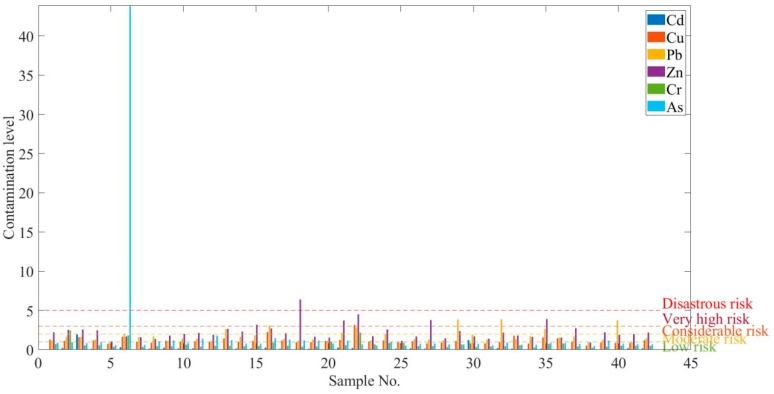
Contamination index of heavy metals.

**Figure 9 ijerph-17-04668-f009:**
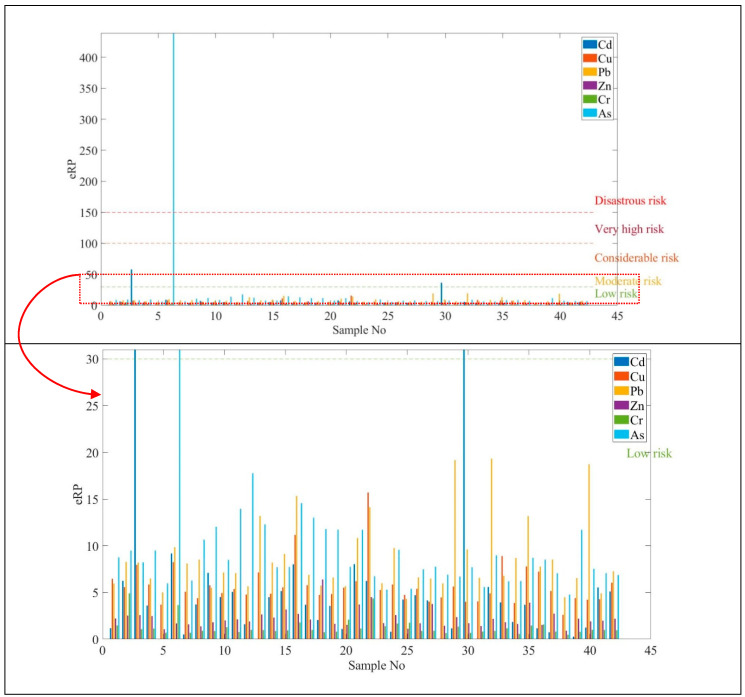
Potential ecological risk index of single element $\left(eRP_{i}\right)$.

**Figure 10 ijerph-17-04668-f010:**
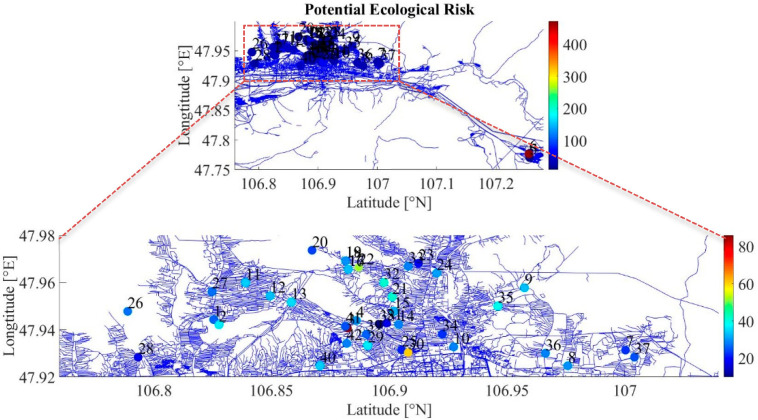
Potential ecological risk to the environment.

**Table 1 ijerph-17-04668-t001:** Procedures of the measurements.

Step 1 Solid	Solid sample	0.05 g	120 °C	48 h (until dry)
	HNO_3_ 60%	3 mL		
	Hydrofluoric 48%	3 mL		
Step 2	Hydrochloric acid 30%	3 mL	120 °C	24 h (until dry)
Step 3 Solution	Extracted solution of HNO_3_ 0.6%	10 mL	Mix motor	24 h
Step 4 Digestion	Extracted solution	10 mL	All soil samples were filtered by 0.20 µm membrane cellulose filter	
Step 5 Dilution (50 times)	Indium standard solution	0.01 g	Total 10 mL diluted solutions were prepared for measurement of the ICP-MS and ICP-OES	
	Extracted solution	0.2 g		
	HNO_3_	9.79 g		

**Table 2 ijerph-17-04668-t002:** Category of EF (enrichment factor) and I_geo_ (geo-accumulation).

EF		I_geo_	
EF < 2	no enrichment	0 < I_geo_	uncontaminated
EF = 3–5	moderate enrichment	0 < I_geo_ < 1	uncontaminated to moderately uncontaminated
EF = 5–10	moderately severe enrichment	1 < I_geo_ < 2	moderately contaminated
EF = 10–25	severe enrichment	2 < I_geo_ < 3	moderately to heavily contaminated
EF = 25–50	very severe enrichment	3 < I_geo_ < 4	heavily contaminated
EF > 50	extremely severe enrichment	4 < I_geo_ < 5	heavily to extremely contaminated
		5 > I_geo_	extremely contaminated

**Table 3 ijerph-17-04668-t003:** RfD (reference dose) and SF (slope factor) [mg/kg day] values used in the present study.

**Elements**		**RfD [mg/kg day]**		
		**Pathways**		
		**Ingestion**	**Dermal**	**Inhalation**
1	Cr	3 × 10^−3 (a)^	3 × 10^−3 (a)^	2.86 × 10^−5 (a)^
2	Pb	1.4 × 10^−3 (a)^	5.25 × 10^−4 (a)^	3.52 × 10^−3 (a)^
3	Zn	3 × 10^−1 (c)^	6 × 10^−2 (c)^	3 × 10^−1 (a)^
4	Cd	3 × 10^−1 (a)^	2.3 × 10^−5 (a)^	1 × 10^−5 (a)^
5	As	1 × 10^−4 (a)^	1.23 × 10^−4 (a)^	1.23 × 10^−4 (a)^
6	Co	2 × 10^−2 (a)^	NA	NA
7	Cu	4 × 10^−2 (a)^	1.2 × 10^−2 (a)^	4 × 10^−2 (a)^
8	Mo	5 × 10^−3 (a)^	NA	NA
**Elements**		**SF [mg/kg day]**		
		**Pathways**		
		**Ingestion**	**Dermal**	**Inhalation**
1	Cr	5 × 10^−1 (a)^	20 ^(a)^	42 ^(a)^
2	Pb	8.5 × 10^−3 (a)^	NA	4.2 × 10^−2 (c)^
3	Cd	NA	NA	6.3 ^(a)^
4	As	1.5 ^(a)^	3.66 ^(a)^	4.3 × 10^−3 (a)^

NA represents data not available. ^a^ USEPA, 2007; ^b^ Huang, 2017; and ^c^ Kamunda, 2016 [25,26,27].

**Table 4 ijerph-17-04668-t004:** Parameters of risk assessment.

Parameters	Adult	Children	Unit	References
ADI, average daily intake	-	-	[mg/kg day]	-
IngR, soil ingestion rate	100	200	[mg/day]	[29]
EF, exposure frequency	350	350	[day/year]	[29]
ED, exposure duration	30	6	[year]	[30]
BW, body weight	70	15	[kg]	[30]
SF, skin area exposed to soil contact	5700	2800	[cm^2^]	[30]
AF, soil to skin adherence factor	0.07	0.2	[kg/cm day]	[30]
ABS, contact factor	0.1	0.1	none	[30]
InhR, inhalation rate	15	10	[m^3^/day]	[31]
PEF, particle emission factor	1.36 × 10^9^	1.36 × 10^9^	[m^3^/kg]	[32]
AT, average time non-carcinogenic	10,950	2190	[days]	[33]
AT, average time carcinogenic	25,550	25,550	[days]	[33]
CR, Conversion factor	1 × 10^−6^	1 × 10^−6^	[mg/kg]	[33]
FE, Dermal exposure ratio	0.61	0.61	-	[30]

**Table 5 ijerph-17-04668-t005:** Background concentration [mg/kg] used to calculate the contamination index.

Cd	Cu	Pb	Zn	Cr	As
1	25	20	60	45	12

**Table 6 ijerph-17-04668-t006:** Category of contamination index.

	$CL_{i}$	Pollution Level
1	$CL_{i}<1$	Non-pollution
2	$1\leq CL_{i}<2$	Low level of pollution
3	$2\leq CL_{i}<3$	Moderate level of pollution
4	$3\leq CL_{i}<5$	Strong level of pollution
5	$CL_{i}>5$	Very strong pollution

**Table 7 ijerph-17-04668-t007:** Relationship among $eRP_{i}$, ERP, and pollution level [34].

$\mathrm{Scope}\;\mathrm{of}\;\mathrm{Potential}\;\mathrm{Ecological}\;\mathrm{Risk}\;\mathrm{of}\;\mathrm{Sin}\mathrm{gle}\;\mathrm{Heavy}\;\mathrm{Element}\;\left(eRP_{i}\right)$	Ecological Risk Level of Single-Factor Pollution	Scope of Potential Ecological Risk (ERP)	General Level of Potential Ecological Risk
$ERP_{i}<40$	Low	PER<150	Low grade
$40<ERP_{i}<80$	Moderate	150<PER<300	Moderate
$80<ERP_{i}<160$	Higher	300<PER<600	Severe
$160<ERP_{i}<320$	High	PER≥600	Serious
$ERP_{i}\geq 320$	Serous	-	-

**Table 8 ijerph-17-04668-t008:** Measurement results of heavy metal concentrations and Mongolian National Standard (MNS) standard.

No.	Elements *N* = 42	Min [mg/kg]	Max [mg/kg]	Average [mg/kg]	Mongolian National Standard (MNS 5850: 2008)		
					Permissible Limit [mg/kg]	Toxic Level [mg/kg]	Dangerous Level [mg/kg]
1	Cr	10.6	110.4	29.0	150	400	1500
2	Co	3.2	13.0	7.6	50	500	1000
3	Cu	13.0	78.6	28.9	100	500	1000
4	Zn	54.8	384.0	135.6	300	600	1000
5	As	5.7	526.8	22.9	6	30	50
6	Mo	8.0	334.8	39.8	5	20	50
7	Se	0	1.5	0.1	10	50	100
8	Cd	0	1.9	0.2	3	10	20
9	Pb	17.3	77.3	34.5	100	500	1200
10	V	38.1	118.2	69.9	150	600	1000
11	Sr	271.2	531.9	353.2	800	3000.0	6000.0
12	Al	45,708.2	68,080.1	57,954.9	-	-	-
13	Ag	0.3	1.9	1.1	-	-	-
14	Kr	0.0	0.7	0.3	-	-	-
15	Rb	57.3	108.8	84.5	-	-	-
16	Cs	2.2	8.2	4.0	-	-	-
17	Ba	447.5	859.8	677.1	-	-	-
18	Bi	0.2	3.1	0.6	-	-	-
19	Th	8.6	34.1	12.7	-	-	-
20	U	1.5	4.0	2.5	-	-	-
21	Ca	1.6	41.5	19.7	-	-	-
22	Fe	11.2	54.0	24.5	-	-	-
23	K	13.0	28.3	22.9	-	-	-
24	Mg	1.1	107.1	30.3	-	-	-
25	Mn	0.1	1.1	0.5	-	-	-
26	Na	2.1	27.9	16.7	-	-	-

**Table 9 ijerph-17-04668-t009:** Results of the hazard index for children.

No.	Elements	Pathways		
		Ingestion	Dermal	Inhalation
1	Cr	2.06 × 10^−2^	1.05 × 10^−1^	4.76 × 10^−4^
2	Co	8.06 × 10^−4^	-	-
3	Cu	1.54 × 10^−3^	2.63 × 10^−2^	3.39 × 10^−7^
4	Zn	9.63 × 10^−4^	2.47 × 10^−2^	2.13 × 10^−7^
5	As	1.63 × 10^−1^	2.03	8.75 × 10^−5^
6	Mo	1.70 × 10^−2^	-	-
7	Cd	3.65 × 10^−2^	8.14 × 10^−2^	8.06 × 10^−6^
8	Pb	2.4 × 10^−3^	7.18 × 10^−1^	4.61 × 10^−6^
	**Total HI**	2.59 × 10^−1^	2.99	5.77 × 10^−4^

**Table 10 ijerph-17-04668-t010:** Results of the hazard index for adults.

No.	Elements	Pathways		
		Ingestion	Dermal	Inhalation
1	Cr	3.55 × 10^−9^	4.60 × 10^−2^	1.53 × 10^−4^
2	Co	1.39 × 10^−10^	-	-
3	Cu	2.65 × 10^−10^	1.15 × 10^−2^	1.09 × 10^−7^
4	Zn	1.66 × 10^−10^	1.08 × 10^−2^	6.83 × 10^−8^
5	As	2.81 × 10^−8^	8.87 × 10^−1^	2.81 × 10^−5^
6	Mo	2.93 × 10^−9^	-	-
7	Cd	6.3 × 10^−11^	3.55 × 10^−2^	2.59 × 10^−6^
8	Pb	9.06 × 10^−9^	3.13 × 10^−1^	1.48 × 10^−6^
	**Total HI**	4.47 × 10^−8^	1.3	1.85 × 10^−4^

**Table 11 ijerph-17-04668-t011:** Results of cancer risk for children.

No.	Pathways	Cr	As	Cd	Pb	Total Cancer Risk
1	Ingestion	2.65 × 10^−6^	6.27 × 10^−6^	-	5.36 × 10^−8^	8.97 × 10^−6^
2	Dermal	5.42 × 10^−4^	7.84 × 10^−5^	-	-	6.2 × 10^−4^
3	Inhalation	2.78 × 10^−11^	2.15 × 10^−7^	1.1 × 10^−12^	3.31 × 10^−8^	2.48 × 10^−7^

**Table 12 ijerph-17-04668-t012:** Result of cancer risk for adults.

No.	Pathways	Cr	As	Cd	Pb	Total Cancer Risk
1	Ingestion	2.87 × 10^−7^	6.27 × 10^−6^	-	5.74 × 10^−9^	9.62 × 10^−7^
2	Dermal	1.18 × 10^−3^	1.71 × 10^−4^	-	-	1.35 × 10^−3^
3	Inhalation	4.47 × 10^−11^	3.45 × 10^−7^	1.76 × 10^−12^	5.32 × 10^−8^	3.98 × 10^−7^

**Table 13 ijerph-17-04668-t013:** Results of contamination index ($CL_{i}$).

Descriptive	Cd	Cu	Pb	Zn	Cr	As
Min	0.11	0.52	0.86	0.91	0.24	0.48
Max	1.93	3.14	3.87	6.4	2.45	43.9
Mean	0.17	1.16	1.73	2.26	0.64	1.91

**Table 14 ijerph-17-04668-t014:** Potential ecological risk index of single element $\left(eRP_{i}\right)$.

Descriptive	Cd	Cu	Pb	Zn	Cr	As
Min	0.11	13.05	17.25	54.82	10.63	5.74
Max	1.93	78.59	77.32	383.96	110.42	526.75
Mean	0.17	28.88	34.50	135.63	28.97	22.90

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
