# Peer review of "Ecological and Human Health Risk Assessment of Heavy Metal Pollution in the Soil of the Ger District in Ulaanbaatar, Mongolia"

_ijerph, 2020, doi:10.3390/ijerph17134668_

Round 1

Reviewer 1 Report

Assessment of heavy metal pollution and human 2 health risk in ger district of Ulaanbaatar, Mongolia

The manuscript is focused on the assessment of elements risk assessment using a range of pathways including: ingestion, dermal contact and inhalation, in Ger district of Ulaanbaatar, Mongolia a zone with a background of studies that demonstrates the presence of elements of concern in high concentrations, using elemental concentration in soil samples. The manuscript contributes with an overall risk assessment that provides acute information about the potential risk for population at the studied zone.

Mayor corrections:

In general redaction must be improved

Line 40-42: Please add some lines that connects coal ash with heavy metal content

Lines 99-104: Before you explains that some authors already asses the concentration of various elements in the zone” and then authors says that the objective of this study es pretty the same, please explain clearly what is the difference with those works and this one clearly. Is the methodology, is the area different?

Figures 2 and 3, figures ate mentioned in discussion but no more information is  available in the manuscript, please add some lines describing the figures, from just take a view of the figures is difficult to understand the difference between them, is there some difference?, even more there is a difference between adults and children’s?.

Minor corrections:

Line 40: “the waste of coal ash has been illegal” you mean “The disposal of coal waste has been illegal”?

Line 46: please replace “risk level related to” for “risk level is related to”

Line 57: please replace “shows the amount of” for “shows that the amount”

Line 62: This is a fact well documented or is a supposition, in the last case you should write “and might be” instead “and be”.

Line 63: Please eliminate “was” after Batkhising.O [6]

Line 78: please repeat the author principal name or write “the same authors” eliminate “researches and et al.”

Line 81: Please specify which limit, is an oral soil ingestion exposure limit?

Line 96: Please replace “the symptom of population” for “the symptoms of population”

Line 123: In this sentence authors says that independent of the values of HI health effects can occur, according to the literature HI <1 indicates that toxic effects are unlikely to occur please correct

Line 141: Please add some reference to justify these criteria

Line 182: Please specify the intention of store the samples for 7 days

Line 185: Please specify why authors chose that particle size

Line 202: Did authors measure a certified standard to assure the recovery of elements in samples, or do you have detection limits and some statistic data that assures the veracity of measurements?

Line 208: Please increase the font number in the legend of the figure 1 for a better appreciation

Line 224: the authors just identify 10 elements, or they chose those presented in the manuscript since those was the most representative, please explain.

Line 228: Please add an explanation of MNS

Line 270: Please modify the Figure 2 caption, “distribution level” is repeated

Line 311: which living conditions and lifestyles authors are referring, please add some examples.

Line 341: Please add a space after “12”

Line 440, 442, 445, please homogenize bibliography use EPA or USEPA in the entire manuscript.

Authors will use “ger” or “Ger” please homogenize

Author Response

Dear reviewer

Thank you very much for the comments and the opportunity to revise our paper which was entitled ‘Risk Assessment of heavy metal pollution in soil and associated human health to residents of ger district in Ulaanbaatar, Mongolia’. The suggestions offered by the reviewers have been immensely helpful, and we also appreciate your insightful comments on revising the introduction, methodology, results and conclusion of the paper.

We have included the reviewer’s comments immediately after this letter and responded to them individually, indicating exactly how we addressed each concern or problem and describing the changes we have made in the correction matrix. The changes are used as the track changes in this paper and the quite revised manuscript is attached to this email.

In addition, we would like to explain the general changes in this manuscript by ourselves which did not mention the reviewers before.       

For example, we have added the new part of potential ecological risk assessment due to assess the contamination index which has been included in the title of our manuscript. Therefore, the title was changed by ‘Ecological and human health risk assessment of heavy metal pollution in the soil of the ger district in Ulaanbaatar, Mongolia’.

We hope the quite revised resubmission paper will better suit the special issue of Assessment of Health-Related Quality of Life (HRQoL) in Public Health in JERPH, but we are happy to consider further revisions and we thank you for your continued interest in our research. 

Sincerely,

Ms. Author, 

â„–

Reviewer comments

Authors correction

Reviewer form 1

1

Line 40: “the waste of coal ash has been illegal” you mean “The disposal of coal waste has been illegal”?

Yes. The disposal of coal waste has been illegal. This sentence has been improved.

For example: Unfortunately, due to lack of management for coal ash disposal in the ger area, residents are disposed of the coal ash waste at illegal and unregulated points such as public waste points and ravines near the ger area.

2

Line 40-42: Please add some lines that connects coal ash with heavy metal content

The connection between coal with heavy metals is information has been mentioned and referenced.

For example, coal ash contains a high concentration of heavy elements comparing with other geological materials. Because the concentration of heavy elements is enriched four to ten times after combustion and it is harmful to human health [8-9].

3

Line 46: please replace “risk level related to” for “risk level is related to”

The sentence is corrected.

4

Line 57: please replace “shows the amount of” for “shows that the amount”

The sentence is corrected.

5

Line 62: This is a fact well documented or is a supposition, in the last case you should write “and might be” instead “and be”.

The sentence is corrected.

6

Line 63: Please eliminate “was” after Batkhising.O [6]

The sentence is corrected.

7

Line 78: please repeat the author principal name or write “the same authors” eliminate “researches and et al.”

The sentence is corrected.

8

Line 81: Please specify which limit, is an oral soil ingestion exposure limit?

The ingestion was estimated by the main contributor to health risk. It has been added to this paragraph.

9

Line 96: Please replace “the symptom of population” for “the symptoms of population”

The sentence is corrected.

10

Lines 99-104: Before you explains that some authors already asses the concentration of various elements in the zone” and then authors says that the objective of this study is pretty the same, please explain clearly what is the difference with those works and this one clearly. Is the methodology, is the area different?

We are explaining about three main different works of this paper. First, Sonomdagva.Ch et al have studied the health risk of heavy metals in soil, including 13 sites in the ger district of Ulaanbaatar. Those 13 samples were could not be the explain a health risk of ger residents. The sample number is low. Second, the preparation of the sampling method is different from the literature. We used the total digestion method for the laboratory. Third, the comparison of literature and the current result was analyzed in this manuscript.

11

Line 123: In this sentence authors says that independent of the values of HI health effects can occur, according to the literature HI <1 indicates that toxic effects are unlikely to occur please correct

Apologies for that incorrect word. It means that if HI >1, it can affect human health. The incorrect word has been changed by ''exceed'' and the sentence is revised. Thank you very much.

12

Line 141: Please add some reference to justify these criteria.

It has been added some reference.

13

Line 182: Please specify the intention of store the samples for 7 days

The obtained samples were stored for a week in a desiccator with air-tight plastic bags and then they used for chemical analysis in the Kanazawa University laboratory.

14

Line 185: Please specify why authors chose that particle size

The same particle size analysis is better to compare former research work. Most of the soil sample analyses were obtained in that size, especially in the case of the Ulaanbaatar.

15

Line 202: Did authors measure a certified standard to assure the recovery of elements in samples, or do you have detection limits and some statistic data that assures the veracity of measurements?          

We have used some references such as De Carvalho et al., 2015, Zemberyova et al., 2006., and Solongo et al., 2018. The detection limit is depending on heavy metals concentration around 50 - 100 ng/kg for low-level elements. The references have been referenced.

16

Line 208: Please increase the font number in the legend of the figure 1 for a better appreciation

The study map has been improved such as the added source of the map has added and font size is increased.

17

Line 224: the authors just identify 10 elements, or they chose those presented in the manuscript since those was the most representative, please explain.

We were selecting those 10 elements due to evaluate the pollution indexes and used in the evaluation of risk assessment. However, we added all of the measured elements in this manuscript during the under reviewer.

18

Line 228: Please add an explanation of MNS

The MNS means Mongolian National Standard. It has been written in the above sentence.

19

Line 270: Please modify the Figure 2 caption, “distribution level” is repeated

The sentence is corrected.

20

Line 311: which living conditions and lifestyles authors are referring, please add some examples.

This sentence has been moved to the conclusion part and explained.

For example, When coal is burnt in the stove of ger dwelling, a lot of ash is produced. A small part of coal ash is dispersed in the air (fly ash) and most of them stay in the stove [39]. After finished the burning process, the remaining ash is collected in a bucket, and mostly thrown into the open street with bare soil. Surprisingly, hot ashes can help to reduce slippery surface and show cover in the open street in the winter period (November till March)[3]. This is the reason why coal ash and its heavy metals fly in the air and accumulate in the soil and dissolve in the water.

21

Line 341: Please add a space after “12”

The sentence is corrected.

22

Figures 2 and 3, figures ate mentioned in discussion but no more information is  available in the manuscript, please add some lines describing the figures, from just take a view of the figures is difficult to understand the difference between them, is there some difference?, even more there is a difference between adults and children’s?.

The figures of carcinogenic and non-carcinogenic are changed by bar chart.

•The 9 of 26 elements were evaluated for human health risk. Because of SF and RD were available. The carcinogenic and noncarcinogenic risks were calculated well-known methods recommended by USEPA. The carcinogenic risk was estimated as the unacceptable range for both adults and children at all of the sample sites. The total noncarcinogenic risk estimated at 3.24E+004 or unacceptable range for children at all of the sample sites. For the adult, only a number of 12 sample sites were estimated at 1.30E+00 or unacceptable range while other sample sites were estimated by no health risk.

23

Line 440, 442, 445, please homogenize bibliography use EPA or USEPA in the entire manuscript.

The correction has been made. For example, all EPA has changed by USEPA. 

24

Authors will use “ger” or “Ger” please homogenize

ger used instead of GER in the entire manuscript.

Reviewer 2 Report

1. The originality of this paper is low. Besides, this manuscript need be major revised. Author(s) may need to strongly address why this study is important to be published. In addition, there are not outstanding merits by compared with the related papers about heavy metal pollution and human health risks.

2. In introduction, the authors are suggested to crisply described that Hg, Co, Se, Ba, Mo are the main source of heavy metals in soil was coal ash in Ger area.

3. In section 2.3 Data collection and sampling method, the authors did not explain why the concentration of 119 soil samples was selected from previous researchers' data.

4. There may have been reasons for determinations of the concentration of target heavy elements by ICP-OES and ICP-MS.

Over-all evaluation
Thus, although the study is basically interesting, the submitted manuscript needs substantial revisions. The manuscript as submitted cannot be recommended for possible publication. The authors are however strongly encouraged to re-submit the revised version of the manuscript, taking the above-mentioned points into consideration when making revisions.

Author Response

Dear reviewer

Thank you very much for the comments and the opportunity to revise our paper which was entitled ‘Risk Assessment of heavy metal pollution in soil and associated human health to residents of ger district in Ulaanbaatar, Mongolia’. The suggestions offered by the reviewers have been immensely helpful, and we also appreciate your insightful comments on revising the introduction, methodology, results and conclusion of the paper.

We have included the reviewer’s comments immediately after this letter and responded to them individually, indicating exactly how we addressed each concern or problem and describing the changes we have made in the correction matrix. The changes are used as the track changes in this paper and the quite revised manuscript is attached to this email.

In addition, we would like to explain the general changes in this manuscript by ourselves which did not mention the reviewers before.       

For example, we have added the new part of potential ecological risk assessment due to assess the contamination index which has been included in the title of our manuscript. Therefore, the title was changed by ‘Ecological and human health risk assessment of heavy metal pollution in the soil of the ger district in Ulaanbaatar, Mongolia’.

We hope the quite revised resubmission paper will better suit the special issue of Assessment of Health-Related Quality of Life (HRQoL) in Public Health in JERPH, but we are happy to consider further revisions and we thank you for your continued interest in our research. 

Sincerely,

Ms. Author, 

Correction matrix

â„–

Reviewers comments

Authors correction

1

The originality of this paper is low. Besides, this manuscript need be major revised. Author(s) may need to strongly address why this study is important to be published. In addition, there are not outstanding merits by compared with the related papers about heavy metal pollution and human health risks.

According to reviewers’ 3 comments, the manuscript is significantly improved. Merit by comparison with the literature about heavy metal pollution and human health risks is highlighted in the last paragraph. The introduction, results, and discussion sections are majorly revised. We are explaining about three main different works of this paper. First, Sonomdagva.Ch et al have studied the health risk of heavy metals in soil, including 13 sites in the ger district of Ulaanbaatar. Those 13 samples were could not be the explain a health risk of ger residents. The sample number is low. Second, the preparation of the sampling method is different from the literature. We used the total digestion method for the laboratory. Third,  the comparison of literature and the current result was analyzed in this manuscript.

2

In introduction, the authors are suggested to crisply described that Hg, Co, Se, Ba, Mo are the main source of heavy metals in soil was coal ash in Ger area.

The source of Hg has been deleted from the introduction part. because it was not measured in ICP-MS. The mentioned it in line between 95 and 100. For the other elements as you mentioned which were not mentioned any heavy metals of source in the ger area. But those elements are constant in coal and coal ash.

3

In section 2.3 Data collection and sampling method, the authors did not explain why the concentration of 119 soil samples was selected from previous researchers' data.

The sample collection has been explained and restructured in methodology. We selected the concentration of heavy elements in soil from literature, especially from ger area due to evaluate the health risk.

4

There may have been reasons for determinations of the concentration of target heavy elements by ICP-OES and ICP-MS.

ICP-MS is one of the best widely used method for identifying the total heavy metals concentration with high sensitivity and wide dynamic range (De Carvalho et al., 2015). ICP-OES is identifying major elements such as Al, Fe, Na, Ca so on. Fe is used in the evaluation of enrichment factors. 

5

Line 208: Please increase the font number in the legend of the figure 1 for a better appreciation.

The figure has been changed. For example, the font number in the legend of the figure has been increased as your suggestion and also the source of the study area has added.

Reviewer 3 Report

The goal of the study is very important and is something that should be studied. It is a priority in terms of public health. However, the manuscript has to be improved. English is very unclear and makes the reading difficult. Also the ideas are not well organized. And the methodology is very unclear. The lab methodology doesn't have any references and QA/QC is not clear how it was assess.

The author has the tendency to mention statements that are not supported by the study. More supportive literature it is definitely needed. It is not clear why the author chooses the elements presented in the study. I recommend that the manuscript has to be rewritten in a more organized and clear way.  I am specifying detailed corrections in the PDF file.  

Author Response

Dear reviewer

We would like to say thank you very much for all of your important comments and the opportunity to revise our paper which was entitled ‘Risk Assessment of heavy metal pollution in soil and associated human health to residents of ger district in Ulaanbaatar, Mongolia’.

The suggestions offered by the reviewers have been hugely helpful, and we also appreciate your insightful comments on revising the introduction, methodology, results, and conclusion of the paper.

We have included the reviewer’s comments immediately after this letter and responded to them individually, indicating exactly how we addressed each concern or problem and describing the changes we have made in the correction matrix. The changes are used as the track changes in this paper and the quite revised manuscript is attached to this email.

In addition, we would like to explain the general changes in this manuscript by ourselves which did not mention the reviewers before.       

For example, we have added the new part of potential ecological risk assessment due to assess the contamination index which has been included in the title of our manuscript. Therefore, the title was changed by ‘Ecological and human health risk assessment of heavy metal pollution in the soil of the ger district in Ulaanbaatar, Mongolia’

We hope the quite revised resubmission paper will better suit the special issue of Assessment of Health-Related Quality of Life (HRQoL) in Public Health in JERPH, but we are happy to consider further revisions and we thank you for your continued interest in our research.

Sincerely,

Ms. Author

  Correction matrix  
â„–  Reviewers comments Authors responses

1

However, the manuscript has to be improved.

According to 3 reviewer’s comments, the manuscript is significantly improved. A merit by compared with the related papers about heavy metal pollution and human health risks is highlighted in the last paragraph of Introduction and Discussion sections. Discussion section is majorly revised.

2

English is very unclear and makes the reading difficult.

Proofed manuscript has been edited by English editing

3

The lab methodology doesn't have any references and QA/QC is not clear how it was assess.

The measurement of the laboratory has been improved and more explained in the methodology section. 

4

It is not clear why the author chooses the elements presented in the study.

We were selecting those 10 elements due to evaluate the pollution indexes and used in the evaluation of risk assessment. However, we are considering all of the measured elements in this manuscript during the under reviewer.

5

Enrichment? what does that mean here?

We assessed the enrichment factor for heavy metals in soil. The enrichment factor is to assess the degree of contamination and potential anthropogenic impact. It has introduced in the methodology part

6

The values were below zero, how is that exceeding permissible limit?

Non-carcinogenic and carcinogenic are acceptable limit are explained differently. For example, If the HI value is less than 1, it does not consider human health effects. However, If the HI value exceeds 1, adverse health effects may occur. For the carcinogenic, If the risk value exceeding 1 × 10-4 is considered to be an unacceptable range, and a value of less than 1 × 10-6 is considered to be safe. Moreover, values between 1 × 10-4 and 1 × 10-6 are considered to lie within an acceptable range. Therefore, both carcinogenic and noncarcinogenic has been exceeding permissible limits or acceptable limit

7

This are not the same results that appears in the table presented this results in the manuscript

The correction has been made.

9

The values were below zero, how is that unacceptable?

Non-carcinogenic and carcinogenic are acceptable limit are explained differently. For example, If the HI value is less than 1, it does not consider human health effects. However, If the HI value exceeds 1, adverse health effects may occur. For the carcinogenic, If the risk value exceeding 1 × 10-4 is considered to be an unacceptable range, and a value of less than 1 × 10-6 is considered to be safe. Moreover, values between 1 × 10-4 and 1 × 10-6 are considered to lie within an acceptable range. Therefore, both carcinogenic and noncarcinogenic has been exceeding permissible limits or acceptable limit

10

The method of analysis is irrelevant information at this point

Analysis method excluded.

11

If this was true, why is necessary to study them?  Is there any symptoms in the population?

This is the result of the previous study. They had not analyzed any symptoms in the population. However, Mongolia health issue is one of main problem, especially cancer health. The health issue of Mongolia was registered in the official site of healthdata.org/mongolia. There are mentioned the most deaths. For example, the causes are ischemic heart disease, stroke, liver cancer, cirrhosis, and stomach cancer, lower respiratory infect. Cr, Arsenic, and Pb can lead for the above health impact.

12

Low again... Again what is the need of studying it?

Previous studies showed that heavy metal concentration in the soil is generally low and not exceeded the MNS level. However, some places have a high concentration of heavy metals in the soil. The number of cases of cancer has been increased in Ulaanbaatar in recent years due to environment and life quality. Therefore, this study focused on heavy metal concentration in the soil that people in UB might be influenced.

13

measure

The sentence is corrected.

14

what symptoms are we talking about?

They did not mention any type of symptoms. However, they mentioned that the heavy metals of soil pollution are necessary to compare with the symptoms of disease in the population. 

15

Very confusing, you mentioned above, different sources in the studies, and here you are saying that the main source is coal ash....

Yes. There are several sources including some factories. However, previous studies concluded that the main source among As is coal ash. Also, we did not explain the source of Arsenic in this paragraph before. We have included the distribution and source of Arsenic as a report of the Urban Environment Agency in Mongolia. For example, The highest concentration of As was determined to waste points in the ger area, main waste disposal area, and auto markets.

16

Very confusing. So the mercury was lower compared other cities, and this mercury come from coal power plans in Ger District.  "The relation of the soil and dust of Hg was well" What does that means?

We have deleted this unnecessary point from in introduction part.

17

The method of analysis is irrelevant information at this point.

The analysis method explained very shortly.

18

Ger is a a name, isn't supposed to be in capital letters all the time? You have it sometimes like this and sometimes the"G' is capital, I suggest you standardize it

It standardized ger entire manuscript. ger used instead of GER in the entire manuscript.

19

I thought it was evaluated by reference number 12. Ulaanbaatar was mentioned among the places studied.

Apologies for that. It has been changed by 'very limit'. Thank you. We are explaining about three main different works of this paper. First, Sonomdagva.Ch et al have studied the health risk of heavy metals in soil, including 13 sites in the ger district of Ulaanbaatar. Those 13 samples were could not be the explain a health risk of ger residents. The sample number is low. Second, the preparation of the sampling method is different from the literature. We used the total digestion method for the laboratory. Third,  the comparison of literature and the current result was analyzed in this manuscript.

20

It seems very desorganaize... it should start for the begining, fist: study site; then sample collections, then, sample analyzes, then, comparison with the literatture and aplication of the Risk methodology

Thank you for your suggestions. The structure of the methodology has been made as your suggestions.

21

I suggest at least name the document instead of just putting the references.

References has been written

22

So here you refer to an addition of all the  riscks of all the elements together?

The risk for each element is calculated by equation 4 that includes all exposure pathways.  According to equation (5), Hazar Index (HI) is expressed by the summation of risk which calculated by equation 4. That means the risk calculated for each element are summed.

23

This two sentences are confusing... Not sure what it means... In my understanding being less than 1, that means there is no risk ar all....and above 1 there is risk

The sentence is corrected.

24

Is this formula means that Fe was determined in all of the samples? This is not mentioned in any part of the manuscript

Yes. We have determined a total of 26 elements on ICP-MS and IC-OES, including Fe which is measured in ICP-OES. It has been explained in section of laboratory in the methodology.

25

This paragraph is extremely confusing, you should describe the methodology you use FIRST. And then mention that a comparison will be made. Also is not clear if the same elements are study in all the studies, and finally.  Also, don't understand this last sentence "evaluated by an acceptable range"?You do an statement in the middle of the methodology.

We have revised this mistaken sentence. The sentence has been restructured. Also, we explained more detail about the selection of sampling locations in the part of the methodology. For example, Although there are a number of studies have performed to identify the concentration of heavy elements in soil samples as mentioned in the introduction, three main references are selected in the present study. Because most of the samples in these three references [6], [12], [14] were collected from the ger area of Ulaanbaatar and showed a high concentration of heavy elements.

26

How were the site for sample collectio selected?  Sre this the same of the previous study? So In the present study 42 samples will be also collected?

A total of 42 soil samples were recollected of this study based on literature. Because most of the samples in these three references [6], [12], [14] were collected from the ger area of Ulaanbaatar and showed a high concentration of heavy elements. The reselected 42 sampling points have been added more explained in the part of the data collection in methodology. Then the comparison of current results and literature has been discussed for the part of the result.

27

Very confusing again!!!! What digestion method did you use? Did you use a mix of all the acid? Does this methodology has a reference?

The total digestion method used in this manuscript. BCR (Community Bureau of Reference) was a modified extraction procedure for the analysis of heavy metals in soil (Zemberyova et al., 2006). The references have been included in reference data. We have improved this sentence. Thank you very much for your comment.

28

This procedure is veeeery confusing, Do you have any reference for it? What about the recovery of the certified reference material? What were your quality controls of this diggestion? At he end the way it is written, you used the diggested soil? The proceedure doesn't make sense... And the step 5and 4 are extremely confusing

The procedure has been improved and used the references. For example, it has been changed according to the following paragraphs. The reagent was in the analytical method and it was divided into five consecutive steps. In the first step solid samples, each 0.05 g was digested completely in 3 mL 60% HNO3 with 3 mL of 48% Hydrofluoric (HF) and heated at 120 0C for 48 hours. The second step was added 3 mL of 30% Hydrochloric acid (HCl) and heated at 120 0C for 24 hours until dry. The third step was extracted in 10 mL of HNO3 0.6% and mixing for 24 hours in the mix-rotor (Solongo et al., 2018).  The obtained extraction solutions were filtered to the I-boy through a 0.20-µm cellulose membrane filter in the fourth step. The final step was 50 times diluted sample by HNO3 0.6% solution for measurement of the IPC-MS and IPC-OES. The procedure matrixes are summarized in Table 1.  BCR (Community Bureau of Reference) was a modified extraction procedure for the analysis of heavy metals in soil (Zemberyova et al., 2006).

29

which elements were measure in which equipment? WHy do you have to measure in both equipments? when you mean reference of multiple standard solution, you mean the calibration curve? Again what Certified reference material did you use and what was the recovery?

The Calibration curve was prepared known concentration of each element based on multi and single standard solution. Low-level heavy metals concentrations for Cr, Co, Cu, As, Se, Cd, Ba, Pb, Mo, and Zn were analyzed using high sensitivity equipment of ICP-MS. Common metal concentrations for Al, Na. K, Ca, Mg, Fe, and Mn was analyzed using high ICP-OES. Fe is used in the evaluation of enrichment factors.

30

What about the location of the possible sources?

Also, highly recommended to use  a reference map where the country is located in the continent, as well as the distict in the continent.

The figure has been changed. For example, the font number in the legend of the figure increased, and also the source of the study area has added. as your suggestion.

31

Since Yurts and Ger are names, shouldn't start with a capital letter?

Yurt is replaced ger.

32

It is highly recommended that the levels found in this study are compared to the other studies mentioned in the introduction to find out if this levels had change through time.

We have compared with literature and current result on concentration of heavy metals in soil as suggestions. The manuscript has been quite improved and expected. Thank you very much all of your important comment. We appreciate you could specify your comments on this point.

33

What units are these? ... It is not correct to put negative values... You don't do that in concentration results. The Se levels are VEERY strange. The negative values are not a concentration what are your LQ and LD?  What is the number of samples?

We checked the Calibration curve of Se and recalculated of concentration. But the result showed the same the negative. It means that Se may much low concentration.

34

I dont think you mention or show the MNS limits in any part of the test.... This has to be shown. Maybe adding a new column in the table with this MNS limits

The last column of the Table 3 shown MNS limit values.

35

It was more than 3 times. in fact was 3.8 times greater. But it is not clear the following sentences for the 41 soil samples. Were all of them the same concentration? this sentences is very confusing. I think what you mean to say  the average level was 3.8 times higher than the MNS limits, all the samples were above the MNS limit and one o them was  23 times higher then the MNS?

The sentences has been improved. For example, The only sample 6th was determined the highest concentration of As which was higher than MNS by 23 times.

36

I don't understand this, very confusing

The whole sentences has been changed and improved

37

Reference is needed for this. Also it need to be specify the dose, as Zn is an element that has function in the organism.  this whole sentence has to be rewritten. It is confusing

The whole paragraph has changed and improved. Also comparison of current result and literature are described by figure.

38

Due to the high levels of Mo and As found in this specific sample, I highly recommend to comment about the location and possible sources that are surrounded.

The whole paragraph has been imroved and exlplained it. Also showed the figure.

39

I will recommend to coment something about Pb, since it is known that can be harm for children in low concentrations.

We understand your concern about harm of Pb to children. We have explained toxic affect for human health at a result of the noncarcinogenic risk assessment part.

For example,

It is acceptable Pb that there is no safe level of lead exposure, particularly for children. Pb can lead brain swelling, kidney disease, cardiovascular problems, nervous system damage, and even death[25].

We appreciate you could specify your comments on this point.

40

this categories of severe, moderately enrichment, should be mentioned in methodology

The enrichment factors classes were explained in methodology part.

41

More explanation is needed. what does the negative values means? the values found were lower than background levels, how do you explain this finding?

Yes. Negative values area showed the lower than background level. It has been included.

42

It is no clear if the concentration used here are the one collected in this stuyd only, or include the concentrations of other studies, since it is mentioned in the methodology that health assessment was going to be analyze in those samples too.

The comparison of literature and current result for concentration of heavy elements in soil has been added the discussions part.

43

The highest risk seems to be in Nalaikb, is there a way to explain this finding?

The concentration of As in the sample No.6 might be measured as extremely high because this sample was collected from the waste area of the old glass factory of the Nalaikh district. The glass is made of a molten mixture of heavy elements including As[38]. The measurement results of present and reference studies were the same at sample No.6

Map has been changed by bar chart.

44

The risk of ALL the elements is  shown in this MAP, with the same color? So all the elements have the same risk??? This is very confusing you should use another way to show this data

After calculating noncarcinogenic and carcinogenic risk posed by the heavy metals, the final result shown at Figure 2 and 3. The Maps has been changed to shown by bar chart for each sites.

45

More explanation of the table is needed for sure....

Is HQ below one, consider a real hazard? How is consider the interaction of elements in HI?

The table has been explained more specifically. Also, the result of all sample sites was shown by the bar chart. Figure 7 a and b have been shown it.

46

the symbols are wrong... the  order is ascendent not descendent

It has been improved.

47

This sentence is too general and inappropriate, all the elements presented HQ much lower than 1, and the author is not talking about this. As it is clafify the only elements that seems to present a risk it is As. Unless the author mention any synegistic effects, that should be mentioned,

The explanation of the result part has been improved. The result of the health risk assessment has been shown in figure 7 a and b.

As illustrated in Figure 7(a), the HI values for children are estimated as higher than the threshold as mentioned in section 2.5. The dotted red line in Figure 7 is representing the threshold or maximum value to accept. The HI values for children estimated as higher than the threshold at all soil samples, the extremely high HI value estimated at sample No.6 as shown in Figure 7(a). On the other hand, the HI values are estimated as lower than the threshold for adults except for samples of No.6, No.16, and No.32. At sample No.6, the highest HI values for adults also estimated the same as children

48

Are this elements poses any Carcinogenic risk at all?

Yes, Those elements consider coal and coal ash. We selected just toxic elements in these risk analyses.

49

Confusing again. Are theelements in an accepatable or un acceptable values?.. In this case isn't the risk above 1 that represent a risk? In this case, was any real carcinogenic risk found in this study?  The author is very unclear when refering to risk. The way it is written , seems that there is a risk, when inf fact there is no-risk. However, it  should be taken in account that if there is active source of the contaminants, this could rise to a level  that could respresent risk in the future and this is why a constant monitoring system should be consider, as well as, a better managing of the waste produced.

Acceptable range means there are may pose to risk it is necessary to make the risk management. Unacceptable range means there is a harmful effect on human health. According to this limits, we can determine to focus on sequences of risk management. For example, which area is the most risk level, which pathway is the most sensitive so on.  Therefore, an unacceptable range and acceptable range are possible to risk area.

We have write to conclusion more specific about it.

For examples

When coal is burnt in the stove of ger dwelling, a lot of ash is produced. A small part of coal ash is dispersed in the air (fly ash) and most of them stay in the stove [39]. After finished the burning process, the remaining ash is collected in a bucket, and mostly thrown into the open street with bare soil . Surprisingly, hot ashes can help to reduce slippery surface and show cover in the open street in the winter period (November till March)[3]. This is the reason why coal ash and its heavy metals fly in the air and accumulate in the soil and dissolve in the water.

This study found that the people residing in the ger district of Ulaanbaatar were found to be at the greatest risk of exposure to heavy metals through the dermal pathway, owing to contaminated media near waste points, ravine, street, and auto services in the ger district. Also, there is one case of ingestion. The domestic cattle of neighbor nomads usually come to seek food from open dump areas. It may one source of the ingestion pathway [40]. If those cattle’s milk and meat sales to consumers, it is necessary to control those cattle's products by the professional inspection agency of the urban city of Ulaanbaatar. 

Therefore, it is necessary to conduct a risk assessment of the drinking water in ger districts, because targeted three (Cr, Mo, and As) elements can potentially affect the groundwater in these ger districts. 

50

Are you talking abou bioacumulation through time? It sill be good to explain how this mechanism works with As. Also Reference should be included.

The conclusion part has been improved and explained how to release the coal ash in street of ger district.

For example:  When coal is burnt in the stove of ger dwelling, a lot of ash is produced. A small part of coal ash is dispersed in the air (fly ash) and most of them stay in the stove [39]. After finished the burning process, the remaining ash is collected in a bucket, and mostly thrown into the open street with bare soil . Surprisingly, hot ashes can help to reduce slippery surface and show cover in the open street in the winter period (November till March)[3]. This is the reason why coal ash and its heavy metals fly in the air and accumulate in the soil and dissolve in the water.

51

This is one of the most important reflexions.

Thank you very much for all of your important comment. 

52

The conclusion should be more specific... mentioning just the elements that are representing a possible risk. The author didn't do any real comparison among this study results and the one from other studies

Conclusion part has been explained more specific each points.

53

most of the elements evaluated didn't present any risk.... So this afirmation is inccorrect. Also the wase point were never  specify geographically, therefore it is impossible to known how close were the sample collection point from them

Yes, only As and Cr could contribute to the estimated carcinogenic risk imposed on the ger district.

Round 2

Reviewer 1 Report

Authors replies successfully to almost all comments I made before. Authors present a new version of the manuscript including ecological risk assessment and several figures were added aswell. Improved manuscript “Ecological and human health risk assessment of heavy metal pollution in the soil of the ger district in Ulaanbaatar, Mongolia”, is focused on the evaluation of human health and ecological risk assessment in the ger district of Ulaanbaatar city in Mongolia.

General comments

English need to be reviewed minutely in all the manuscript.

The use of the term “heavy elements” is used commonly for metals with high density, however authors are assessing metalloids such as As, other terms more inclusive can be used; trace elements or potentially toxic elements.

The section of Results and discussion, seem to be just a description of the results and a minimum of discussion is presented.

Format of some presented figures need to be improved.

Author Response

Dear reviewer

We would like to thank you for the letter and the opportunity to resubmit a revised copy of this manuscript. We would also like to take this opportunity to express our thanks to the reviewers for the positive feedback and helpful comments for correction or modification.

The changes are used as the track changes in the resubmission manuscript. Also, some additional paragraphs have been added which marked by red color in the revised manuscript. The revised manuscript is attached to this email. We have included the reviewer’s comments immediately after this letter and responded to them individually, indicating exactly how we addressed each concern or problem and describing the changes we have made.

We hope the revised resubmission paper will better suit the special issue of Assessment of Health-Related Quality of Life (HRQoL) in Public Health in JERPH, but we are happy to consider further revisions and we thank you for your continued interest in our research.

Please check the attached files.

Best regards

Author,

Correction matrix

Reviewer 1

1

Line 7: “Department of Transdisciplinary Science and Engineering” is repeated?

The repeated is deleted.

2

Lines 36-37: The sentence is confusing, please rewrite it. I suggest “Igeo and EF indexes were evaluated, indicating the anthropogenic source of As and Mo”.

The sentence has been revised.

For examples,

Soil contamination was assessed by pollution indexes such us Igeo and EF. Mo and As were the most enriched elements comparing with other elements.

3

Line 38: The sentence is confusing, please rewrite it. I suggest “Other elements were assessed but no effects of enrichment or contaminations were obtained”.

The sentence has been revised, as we mentioned for the previous response.

4

Lines 38-41: The sentence is confusing, please rewrite it. I suggest “Carcinogenic and noncarcinogenic risk for child exceeds the permissible limits, and for adults only 12 of 42 sampling points exceeds the limit of noncarcinogenic effects.”

The sentence is corrected, as you suggested.

5

Lines 40-41: It is difficult to understand the aim of the sentence “The Pb, Cr and As could pose serious concern regarding the potential occurrence of health hazards”, please clarify it or eliminate it.

The sentence has been deleted, as you suggested

6

Lines 41-44: The sentence is confusing, please rewrite it. I suggest “According to results of ecological risk assessment, Zn and Pb showed from moderate to considerable contamination indexes and high toxicity values for ecological risk of single element”

The sentence is corrected, as you suggested.

7

Line 45: Keyword “heavy elements” can be replaced for “trace elements” or other more common term.

The all of heavy element names have been changed by trace elements, as you suggested.

8

Line 51: Please replace “are” with “were, since authors are writing about data in the previous year.

The sentence is corrected.

9

Lines 51-53: Please join the sentences is no necessary separate them since are in the same context.

The sentence has been improved.

10

Line 53: Please replace “were” with “was”, since the subject is coal.

The sentence is corrected.

11

Line 56: Please replace “residents are disposed of the coal ash waste at” with “residents are disposing coal ash in”.

The sentence is corrected.

12

Line 61: Please replace “human” with “humans”.

The sentence is corrected.

13

Line 89: Please replace “method” with “technique”, since AAS is not a methodology but an analysis technique.

The sentence is corrected.

14

Line 90: Please replace “as” for “is”.

The sentence is corrected.

15

Line 93: There is a “.” after Sonomdagva, is this a mistake?

The surname first word had written after Sonomdagva. Therefore the dot is not a mistake.

16

Line 96: “Signification” what authors are trying say?

It means there is no risk. This sentence has been improved. For example, the ''significant'' word has been deleted.

17

Line 101: please eliminate “possible” it seems like the authors are possible performing a risk assessment.

The sentence is corrected.

18

Figure 1: Legend in the upper left map is unreadable, please fix it or eliminate it. Is there a difference between color of Middle ger area and districts? please fix it.

The map has been fixed, such as the middle ger area and peri-urban ger area has been consolidated in the map and placed name in the legend is eliminated.

19

Figure 1 caption: Please replace “samples” with “sampling points”, since you are showing the sampling points and not the samples taken.

The sentence is corrected.

20

Line 124: Recorded? Did authors mean collected?

The authors have been registered the coordinate of sampling points by the GPS, or the Global Positioning System. Therefore, we used the ''record'' word.

21

Line 132: This information is already mentioned in the previous paragraph

The sentence is revised.

22

Lines 134-138: please check the sentence is very hard to understand and English need to be reviewed.

The sentence has been improved by the English editing company.

23

Table 1: What authors mean with “Power W”. Extracted solution is a reagent? Is all soil samples a measure of time?. Please fix it.

This unnecessary section has been deleted.

24

Line 227: Authors need to add a discussion for figure 3.

We have deleted figure 3. Because most of the sampling points were estimated exceed than Mongolian National Standards for As. Therefore, the distribution of the map unnecessary and it is deleted from this manuscript. I apologize for the inconvenience. 

25

Figur3 3: Is difficult to link the distribution map of As with the studied are only vi taking a view of coordinates, authors need to think in other way to present it.

Thank you for your comment. It has been deleted.   

26

Line 230: Please add “been” after have

The sentence is corrected.

27

Line 237: Please replace “method” with “technique”

The sentence is corrected.

28

Figure 3: Please explain in the text of the manuscript the figure and how authors can interpret them.

Thank you for your comment. It has been deleted.   

29

Figure 7: Please broke the axis for sample 6 in order to improve visualization of other columns

We apologize for the inconvenience. We understand your situation. But may we stay no change? If it is broke the axis for sample 6, it would be more difficult to understand. We made magnified it possibility other maps.

30

Line 344: Is no a lack of SF parameters, is because not for all elements exist evidence of carcinogenic effects.

It was already mentioned in the methodology part. Therefore we have been deleted. Thank you for your comment.

31

Figure 9: Please broke the axis for sample 6 (As) in order to improve visualization of other columns and try to add all the labels for samples.

We apologize for the inconvenience. We understand your situation. But may we stay no change? If it is broke the axis for sample 6, it would be more difficult to understand. We made magnified it possibility other maps

32

Figure 10: Please broke the axis for sample 6 (As) in order to improve visualization of other columns and try to add all the labels for samples. After this you can eliminate some of the figures presented.

We apologize for the inconvenience. We understand your situation. But may we stay no change? If it is broke the axis for sample 6, it would be more difficult to understand. We made magnified it possibility other maps

33

Conclusions: Please rewrite conclusions, some of the paragraphs are not conclusions but a resume of methodology, others are discussion that can be moved to discussion section.

The conclusion part has been revised such as deleted some unnecessary parts.

For example,   

34

English need to be revised minutely in all the manuscript. The use of the term “heavy elements” is used commonly for metals with high density, however authors are assessing metalloids such as As, other terms more inclusive can be used; trace elements or potentially toxic elements. The section of Results and discussion, seem to be just a description of the results and a minimum of discussion is presented.

The English editing company has been checked and revised.

35

Major revisions

Line 157: is there some information regarding quality analysis, standards such a certified soil material and recovery data?

The reference data has been cited for part of the enrichment factor and geo accumulation index.

Reviewer 2 Report

All comments are replied and addressed.

Author Response

Dear reviewer

We would like to thank you for the letter and the opportunity to resubmit a revised copy of this manuscript. We would also like to take this opportunity to express our thanks to the reviewers for the positive feedback and helpful comments for correction or modification.

Best regards

Author